# Adiponectin and Its Mimics on Skeletal Muscle: Insulin Sensitizers, Fat Burners, Exercise Mimickers, Muscling Pills … or Everything Together?

**DOI:** 10.3390/ijms21072620

**Published:** 2020-04-09

**Authors:** Michel Abou-Samra, Camille M. Selvais, Nicolas Dubuisson, Sonia M. Brichard

**Affiliations:** Endocrinology, Diabetes and Nutrition Unit, Institute of Experimental and Clinical Research, Medical Sector, Université Catholique de Louvain, 1200 Brussels, Belgium; michel.abousamra@uclouvain.be (M.A.-S.); camille.selvais@uclouvain.be (C.M.S.); nicolas.j.dubuisson@uclouvain.be (N.D.)

**Keywords:** Adiponectin, skeletal muscle, AMPK, PGC-1α, insulin signaling, obesity, inflammation, oxidative stress, regeneration, exercise, dystrophinopathies, AdipoRon

## Abstract

Adiponectin (ApN) is a hormone abundantly secreted by adipocytes and it is known to be tightly linked to the metabolic syndrome. It promotes insulin-sensitizing, fat-burning, and anti-atherosclerotic actions, thereby effectively counteracting several metabolic disorders, including type 2 diabetes, obesity, and cardiovascular diseases. ApN is also known today to possess powerful anti-inflammatory/oxidative and pro-myogenic effects on skeletal muscles exposed to acute or chronic inflammation and injury, mainly through AdipoR1 (ApN specific muscle receptor) and AMP-activated protein kinase (AMPK) pathway, but also via T-cadherin. In this review, we will report all the beneficial and protective properties that ApN can exert, specifically on the skeletal muscle as a target tissue. We will highlight its effects and mechanisms of action, first in healthy skeletal muscle including exercised muscle, and second in diseased muscle from a variety of pathological conditions. In the end, we will go over some of AdipoRs agonists that can be easily produced and administered, and which can greatly mimic ApN. These interesting and newly identified molecules could pave the way towards future therapeutic approaches to potentially prevent or combat not only skeletal muscle disorders but also a plethora of other diseases with sterile inflammation or metabolic dysfunction.

## 1. Introduction

Adiponectin (ApN) is a hormone abundantly secreted by adipocytes; it is found in plasma at high concentrations (_~_ μg/mL; at least 3 orders of magnitude higher than other hormones). Plasma ApN levels are decreased in obesity and in patients meeting the criteria for the metabolic syndrome. Yet, ApN could thwart simultaneously several facets of this syndrome by its insulin-sensitizing, fat-burning, and anti-inflammatory/oxidative properties [1].

ApN is synthesized as a monomeric peptide composed by a collagenous stalk and a C-terminal globular domain [2,3,4]. ApN monomers can then assemble into complexes of up to 18 mers (Figure 1). Thus, ApN circulates in the bloodstream as low-molecular-weight form (LMW; trimer), middle-molecular-weight form (MMW, hexamer) and high-molecular-weight form (HMW, 12–18 mers) [5]. Importantly, not only the total levels of ApN, but also the complex distribution of its forms contribute to the biological effects of the hormone. HMW ApN elicits the most potent insulin-sensitizing effects [6]. Finally, ApN could also be cleaved into a smaller globular fragment (gApN) by leukocyte elastases secreted from activated monocytes and neutrophils [7,8].

AdipoR1 and AdipoR2 are the major adiponectin receptors in vivo [9]. They contain seven transmembrane domains but are structurally and functionally distinct from G-protein coupled receptors. AdipoR1 is mainly expressed in skeletal muscle, whereas AdipoR2 is predominantly expressed in the liver [10]. AdipoR1 is tightly linked to activation of AMP-activated protein kinase (AMPK) pathways, while AdipoR2 seems to be associated with activation of peroxisome proliferator-activated receptor (PPAR)-α pathways [9,11]. AdipoR1 is a high-affinity receptor for gApN but can also bind full-length ApN. AdipoR2 shows intermediate affinity for both the globular and the full-length forms [10]. T-cadherin is an additional binding partner for MMW and HMW isoforms of ApN [12]. T-cadherin is a glycosylphosphatidylinositol (GPI)-anchored cadherin, which lacks transmembrane and cytosolic domains and is expressed in heart, aorta, and skeletal muscle. In these tissues, ApN could stimulate exosome production to potentially enhance whole-body metabolism [13]. This receptor further mediates cardiovascular protection [14,15] and skeletal muscle regeneration by HMW ApN [16].

ApN is a critical messenger for the crosstalk between adipose tissue and other metabolic related organs [17] (Figure 1). In the liver, ApN stimulates fatty-acid oxidation [18] and reduces glucose production, thereby exerting an insulin-sensitizing effect on this organ [19]. In adipose tissue, ApN exerts insulin-sensitizing properties through its anti-inflammatory action combined to an increase of glucose uptake [20,21,22]. Targeted overexpression of native ApN into adipose tissue of transgenic mice resulted in adipose tissue remodeling, characterized by younger and smaller adipocytes exhibiting some degree of beiging [21]. In the pancreas, ApN increases glucose-induced insulin secretion through activation of fatty-acid oxidation [23] and prevention of beta cell apoptosis [24]. The effects of ApN on skeletal muscle are summarized in Figure 1 and will be extensively described later (Section 3). ApN also exerts anti-atherosclerotic and cardio-protective effects by inducing the production of nitric oxide (NO) in endothelial cells [25], and by acting as anti-inflammatory and anti-apoptotic agent on the cardiovascular system [26]. In addition, ApN is able to regulate food intake, energy expenditure, and lipid and glucose metabolism by acting on the hypothalamus [27,28]. It also exerts anti-depressive effects by increasing hippocampal neurogenesis and reducing neuroinflammation [29,30,31]. ApN further exhibits renoprotective effects against oxidative stress and apoptosis [32]. Eventually, ApN could protect against cancer progression in most obesity-related cancer types, such as colorectal and breast cancers, through its abilities to induce apoptosis and to limit cell proliferation and angiogenesis [33].

## 2. Adiponectin Production by Skeletal Muscle

Skeletal muscle has been identified as an endocrine organ that has the capacity to produce and secrete myokines. The muscle secretome consists of several hundred cytokines or peptides, which may exert local effects (via an autocrine or paracrine mode) or systemic effects (via an endocrine mode). These myokines mediate immune and metabolic responses [34,35].

Besides being an adipokine, ApN can also be a true myokine [36,37]. Although it is mainly secreted by adipocytes under normal conditions, ApN may be produced in skeletal muscle of mice submitted to stressful challenges. Thus, acute (lipopolysaccharides (LPS) injection) or low-grade inflammation (genetic or nutritional induced obesity) enhanced ApN expression in muscle of challenged mice [36,38,39,40]. ApN up-regulation was also reproduced in cultured murine or human myotubes in response to pro-inflammatory cytokines, reactive oxygen species (ROS) inducers or triglycerides [36,38]. This led to the hypothesis that muscle ApN could be a local protective mechanism to counteract excessive inflammatory reaction, ectopic fat deposit, and oxidative damage. This paradigm was then validated in ApN-knockout (ApN-KO) mice submitted to inflammatory challenge: muscle electro-transfer of the *ApN* gene actually corrected local inflammation, oxidative stress and apoptosis [39,40]. Muscle ApN may also exert locally pro-myogenic properties, as described later (see Section 3.3).

## 3. Adiponectin Properties on Healthy Skeletal Muscle

Skeletal muscle is an important target where ApN regulates energy metabolism, counterbalances inflammation and oxidative stress, and improves tissue regeneration [5]. As ApN is not yet available for therapeutic use in humans, its properties discussed hereafter have been studied in vivo only in rodents receiving/overexpressing the hormone or conversely exhibiting ApN/AdipoR deficiency. However, most mouse data have been also confirmed in vitro in human muscle cells.

### 3.1. Fuel Partitioning and Metabolic Effects

#### 3.1.1. Fatty-Acid Oxidation

ApN exerts its metabolic effects on the skeletal muscle mainly by activating the AMPK signaling pathway (Figure 2). AMPK is a key sensor of cellular energy status, which plays a critical role in systemic energy balance. It is activated after phosphorylation by distinct upstream kinases such as liver kinase B1 (LKB1) or Ca^2+^/calmodulin-dependent protein kinase kinase beta (CaMKKβ).

When ApN links to AdipoR1, it induces the binding of the adaptor protein APPL1, which functions as an anchoring protein tethering LKB1 [41]. ApN may also induce Ca^2+^ influx, thereby activating CaMKKβ, which leads to increased expression of peroxisome proliferator-activated receptor γ coactivator-1α (PGC-1α), a key player in mitochondrial function and biogenesis [42,43]. Both LKB1 and CaMKKβ fully activate AMPK. Next, P-AMPK phosphorylates PGC-1α and indirectly increases the expression of a deacetylase, sirtuin-1 (SIRT1). SIRT1 then completely activates PGC-1α by deacetylation. Thus, ApN stimulates both expression and activation of PGC-1α. In turn, PGC-1α increases the activity of several transcriptions factors (TF), thus stimulating the expression of target genes leading to an increase of mitochondrial biogenesis and a shift towards an oxidative metabolism [11] (Figure 2).

AMPK is also implicated in the sequential activation of p38 mitogen-activated protein kinase (MAPK) and PPARα [44] (Figure 2). By this mechanism, ApN increases the transcriptional activity of PPARα and the expression of its target genes, implicated in fatty-acid transport into the mitochondria (carnitine palmitoyltransferase 1, *CPT1*) and oxidation (Acyl-CoA Oxidase, *ACO*) [44] (Figure 2). Eventually, AMPK further increases fatty-acid oxidation by inhibiting acetyl-CoA carboxylase 2 (ACC2) through phosphorylation (Figure 2). ACC is the enzyme that catalyzes the committed step in fatty-acid synthesis. A decrease in ACC activity reduces the levels of intracellular malonyl-CoA, an allosteric inhibitor of CPT1. Hence, CPT1 can assume the transfer of long-chain fatty acids from the cytosol into the mitochondria, where they are oxidized [45,46].

#### 3.1.2. Insulin-Sensitizing Action

Another core protective effect of ApN against the metabolic syndrome is due to its insulin-sensitizing action. High-fat diet-induced obesity leads to reduced expression of ApN, thereby causing insulin resistance [47]. Several in vivo studies have demonstrated that ApN can greatly improve insulin signaling. Administration of ApN lowered blood sugar and reduced insulin resistance in mice under a high-fat diet (HFD) [47]. Similarly, ApN-transgenic (ApN-Tg) mice are more sensitive to insulin [21], while ApN-KO mice are more resistant [48]. Finally, ApN-Tg *ob/ob* mice (obese mice overexpressing ApN) showed partial attenuation of insulin resistance and diabetes [49].

In the skeletal muscle, ApN can regulate insulin sensitivity through various processes (Figure 2). First, ApN activates AMPK signaling and P-AMPK inhibits p70 ribosomal S6 kinase 1 (p70S6K1), an enzyme that inactivates the insulin receptor substrate-1 (IRS-1) through phosphorylation on serine residues, thereby blocking the insulin signaling cascade [50]. Second, activation of PGC-1α enhances the expression of several oxidative-stress detoxification enzymes and molecules involved in fatty-acid oxidation [43,51]. As both oxidative stress and increased triglyceride content are linked to insulin resistance by inhibitory phosphorylation in IRS-1, ApN further acts as a sensitizing agent [52,53]. Third, ApN induces the translocation of GLUT4 to the plasma membrane, thereby promoting glucose uptake [10,54]. This translocation results from stimulating p38 MAPK via either AMPK [10,44] or interaction with APPL1 [55]. Alternatively, interaction between APPL1 and Rab5 (a small GTPase) may also contribute. Eventually, APPL1 promotes IRS-1 binding to the insulin receptor and enhances insulin signaling [55]. These crosstalks between ApN and insulin signaling pathways underlie the insulin-sensitizing effects of the adipokine.

### 3.2. Control of Inflammation and Oxidative Stress

We previously showed that muscles of ApN-deficient mice displayed higher susceptibility to oxidative stress, inflammation, and apoptosis; all these abnormalities were exacerbated by acute (LPS) or chronic (obesogenic diet) inflammatory challenge and corrected by local electro-transfer of the *ApN* gene [39,40]. Mice with muscle-specific disruption of AdipoR1 also exhibited a local decrease in oxidative-stress-detoxifying enzymes [11]. Moreover, ApN disclosed strong anti-inflammatory properties in human myotubes when submitted to an inflammatory challenge [56,57]. ApN could also put a brake on local inflammation by functioning as a direct regulator of macrophage phenotype favoring the switch from a pro-inflammatory M1-like state to an anti-inflammatory M2-like state, as shown in vivo and in vitro [58]. M2 cells are known to secrete the anti-inflammatory cytokine IL-10 and down-regulate the production of pro-inflammatory cytokines [58]. Thus, ApN appears to be necessary to control inflammation and oxidative stress in muscle.

Mechanistically, the anti-inflammatory effects of ApN in human “healthy” myotubes were linked to activation of the AdipoR1-AMPK-SIRT1-PGC-1α pathway [56,57]. The AMPK pathway is known to inhibit inflammation and oxidative stress by repressing the activity of the transcription factor, nuclear factor kappa B (NF-κB), a key inducer of inflammatory responses [59]. Indeed, both PGC-1α and SIRT1 could repress NF-κB [60,61] (Figure 2). Moreover, we have recently identified a strong microRNA candidate, mir-711 for mediating the anti-inflammatory action of ApN on mouse and human skeletal muscles [62]. ApN up-regulated the expression of miR-711, which, in turn, also repressed NF-κB as well as the inflammasome complex [62,63]. Eventually, as mentioned earlier, PGC-1α is also a regulator of cellular oxidant–antioxidant homeostasis by stimulating gene expression of several antioxidant enzymes such as superoxide dismutase-2, catalase, and glutathione peroxidase [43].

### 3.3. Pro-Myogenic Effects

After acute (reversible) injury, skeletal muscle has a remarkable capacity to initiate a rapid and extensive regeneration, which is always associated with an inflammatory reaction that varies in intensity [64]. Evidence points out that a controlled and efficient inflammation is necessary for an optimal muscle recovery [65]. Regeneration of skeletal muscle also depends on muscle stem cells, named satellite cells, which are kept in a quiescent state in a healthy muscle [64].

After injury, immune cells, including M1 macrophages, are recruited to the lesion site where they produce and release elastases that cleave native ApN into its globular ApN form (gApN) [8]. As gApN expression is low in the bloodstream, it is likely that most gApN is locally produced by muscle (satellite cells, differentiating myoblasts, ..) and acts in an autocrine or paracrine manner as a tissue regenerating hormone [66]. Both M1 macrophages and gApN play essential roles in the activation of satellite cells and the initiation of muscle regeneration [65,66]. Once activated, satellite cells will migrate, proliferate and give rise to myoblasts that differentiate into myotubes, which then fuse to restore damaged fibers or generate new one [64] (Figure 3).

T-cadherin seems to be essential for facilitating ApN-mediated muscle regeneration. It has recently been shown that ApN overexpression in mice decreased necrotic region and increased regenerating myofibers after muscle injury. Yet, these effects were abolished in T-cadherin knockout mice. Moreover, ApN appears to regulate several steps of the repair process by activating different signaling pathways, including p38MAPK and AMPK [66], and different myogenic TF such as Myf5, MyoD, Myogenin, and Mrf4 [56]. First, up-regulation of Myf5 by ApN in skeletal muscle [56] allows the activation of satellite cells and their commitment to regenerate the skeletal muscle [67]. gApN also elicits a specific motility program in satellite cells by allowing them to produce specific metalloproteases, which can degrade the extracellular matrix surrounding muscle fibers, thus driving a proteolytic migration to reach the injury site [68]. Second, gApN induces the expression of MyoD that allows myoblast proliferation and differentiation [56,69,70]. gApN also stimulates autophagy in myoblasts via an AMPK-dependent mechanism, promoting their survival and inhibiting their apoptosis [71]. Third, ApN can coordinate the last step of skeletal muscle regeneration. Indeed, it activates the expression of two key factors of muscle differentiation, Myogenin and Mrf4 [56], which are essential for provoking cell fusion into multinucleated myotubes [67] (Figure 3).

Muscle regeneration is a very complex process involving sometimes non-resident progenitor cells with myogenic properties, such as mesoangioblasts. Upon injury, these cells are attracted to the site of damage where they can differentiate or fuse with pre-existing myofibers [72]. gApN can serve as a positive regulator in mesoangioblasts homing to injured or diseased muscles of mice , thus driving their commitment towards the skeletal muscle lineage [73] (Figure 3).

There is an interplay among the 3 major functions of ApN in skeletal muscle. ApN plays a crucial role in coordinating resolution of inflammation and energy metabolism for an efficient muscle regeneration. ApN promotes the conversion of M1 to M2 macrophages, thereby favoring the resolution of inflammation [58]. Meanwhile, ApN enhances PGC-1α expression and activity, which promotes oxidative metabolism to respond to the high energy demand for sustaining tissue-healing and regeneration [74]. Thus, controlled accumulation of immune cells and increased energy metabolism are essential for appropriate muscle tissue remodeling [65] (Figure 3).

## 4. Adiponectin and Exercise

Exercise training is well-known to improve insulin sensitivity and lipid profile, and to potentially reduce fat mass. Circulating ApN is decreased in obese subjects and in those with the metabolic syndrome. Because exercise and ApN share similar signaling pathways [75,76] and the potential of counteracting several facets of this metabolic syndrome, numerous studies attempted to establish relationships between exercise and ApN parameters. Yet, interpreting these data is challenging because of differences in exercise type and program, individual characteristics of the participants and the pathological context. Moreover, the oligomeric distribution of circulating adiponectin was not always taken into account. Finally, attributing the beneficial effects observed to a reduced fat mass or to the exercise *per se* remains challenging [77].

### 4.1. Effects of Exercise on ApN Parameters

Effects of acute exercise on ApN parameters are inconsistent, if any [77,78]. We will focus on chronic exercise training and its effects on circulating ApN and muscle AdipoRs expression.

Chronic exercise training imposed to rodents for 4 to 16 weeks increased circulating ApN levels in some [79,80], but not all [81], studies. The rise in total plasma ApN was observed at moderate to high aerobic exercise intensity and was associated with a rise in HMW ApN, the metabolically active form of the hormone [79]. These rises correlated with insulin sensitivity. Likewise, chronic exercise (4–12 weeks) induced a rise in total plasma ApN or in the ratio between HMW/total ApN in some human cohorts: subjects with normal glucose tolerance, impaired glucose tolerance or type 2 diabetes, and in patients with hypertension [82,83,84]. Once again, these rises correlated with improved insulin sensitivity [82,83]. In parallel to imposed bouts of exercise training, circulating ApN was linked to physical activity behavior in large human cross-sectional studies, where greater physical activity was generally associated with higher ApN levels [77,85,86]. This relationship between exercise training/physical activity and ApN levels was more deeply discussed recently by others [77].

Chronic exercise training also induced an increase in muscle AdipoR1 expression in rodents [80,87,88] and in humans [82,83]. Usually, muscle ApN levels were unmodified, but this may depend on the studied muscle type [89].

Most of the positive changes induced by chronic physical training (circulating total ApN, oligomeric distribution and enhanced membrane receptor expression) correlated with improved insulin sensitivity. Yet, it cannot be ruled out that weight loss *per se* could also contribute to this improvement.

### 4.2. Contribution of ApN to Exercise-Induced Improvement in Metabolic Fitness

More direct evidence of ApN/AdipoR1 contribution to improved metabolic fitness comes from a study in aged mice. Exercise training for 4 months attenuated the decline in muscle regeneration and performance in SAMP10 mice, a senescence-accelerated mouse model. Muscle benefits induced by chronic exercise were diminished by ApN or AdipoR1 blockade. This blockade was achieved by using specific antibodies directed against either ApN or AdipoR1 for 2 months. These data indicate *a contrario* that exercise training may exert beneficial effects on muscle through ApN-AdipoR1 axis [80]. Moreover, muscle of ApN-KO mice showed contractile dysfunction in situ [37]. Likewise, mice with muscle-specific disruption of AdipoR1 exhibited decreased PGC-1α expression and activity, decreased mitochondrial content and function, as well as decreased oxidative type 1 myofibers and oxidative-stress detoxifying enzymes. All these abnormalities were associated with decreased insulin sensitivity in muscle and reduced endurance capacity as assessed by treadmill running [11]. In humans, ApN up-regulation may be involved in the insulin-sensitizing effects of exercise and the protection against inflammation, muscle damage and excessive ROS production [90]. Thus, ApN is required for mediating the effects of physical exercise.

### 4.3. Contribution of ApN to Exercise-Induced Improvement of Mental Health

Besides its beneficial effects on metabolic and cardiovascular disorders, physical exercise may also contribute to treat depressive disorders. The anti-depressive effect of exercise is mediated in part through enhanced hippocampal neurogenesis and increased dendritic plasticity in rodents [91].

Like exercise, ApN, appears to exert neuroprotective effects in the central nervous system, in addition to its peripheral effects [30]. This may be explained by the fact that trimeric and hexameric forms of ApN may pass the blood–brain barrier and that ApN receptors are expressed in the brain [92,93].

It has been shown in mice that ApN plays a significant role in mediating the effects of exercise on hippocampal neurogenesis and depression, possibly by activating the AdipoR1-AMPK signaling pathway. Thus, running induces a rise in hippocampal ApN levels, while running-exerted hippocampal neurogenesis and antidepressant effects were diminished in ApN-KO mice [30].

## 5. Adiponectin Properties on Diseased Skeletal Muscle

### 5.1. Beneficial Effects of Adiponectin on the Dystrophic Muscle

Muscle may be a site of a severe form of chronic inflammation, which in turn plays a central role in the development or evolution of several myopathies [94,95]. Because of its anti-inflammatory and pro-myogenic properties [96], ApN could offer interesting therapeutic prospects for managing muscle diseases. We will mostly focus on Duchenne muscular dystrophy (DMD), the most frequently inherited human myopathy and the most devastating type of muscular dystrophy [97]. Next, we will briefly tackle other muscular dystrophies.

#### 5.1.1. Duchenne Muscular Dystrophy

DMD is caused by a defective gene that encodes for dystrophin, a key scaffolding protein that provides structural stability and integrity to muscle fiber membrane [98]. Dystrophin-deficient fibers are highly susceptible to injury, resulting in endless cycles of muscle necrosis and repair that lead to fibrosis and weakness [97,98]. Although *dystrophin* gene mutations represent the primary cause of DMD, it is the secondary processes involving persistent inflammation and subsequent impaired regeneration that likely exacerbate disease progression [94]. DMD remains a rapidly progressive and lethal disorder, where patients are typically wheelchair bound by 8–14 years of age and die from cardiac or respiratory failure during their third decade [99].

We and others have shown that circulating levels of ApN are greatly decreased in mdx mice (mouse model of DMD) [56,100], an observation confirmed in human patients [101]. Moreover, myotubes from DMD patients were unable to produce ApN for local protection [57], unlike non-dystrophic myotubes challenged by inflammation [36]. Thus, there could be a rationale to therapeutically correcting the low levels of ApN in dystrophic patients.

Replenishment of ApN by transgenesis or gene electro-transfer mitigated the dystrophic phenotype in mdx mice [56,102]. Indeed, when we generated transgenic mdx mice overexpressing ApN (mdx-ApN mice), we showed that ApN can act as preventive agent and delay disease progression [56]. First, correction of ApN levels was crucial to counterbalance inflammation and oxidative stress in skeletal muscle. ApN drastically reduced the expression of pro-inflammatory cytokines (TNF-α and IL-1β), and of oxidative-stress markers (peroxiredoxin, an antioxidant enzyme, and 4-hydroxynonenal, a lipid peroxidation product). Dystrophic muscles from mdx-ApN mice showed significant increased expression of the anti-inflammatory cytokine (IL-10), and were protected from T-lymphocytes and M1 (pro-inflammatory) macrophages infiltration [56]. M1 macrophages actually switched into M2 (tissue-healing) state. Second, ApN positively boosted the skeletal myogenic program. Skeletal muscles of mdx-ApN mice displayed higher expression of two major factors of muscle proliferation (Myf5 and MyoD), and two major factors of muscle differentiation (Myogenin and Mrf4) [56,67]. Thus, ApN stimulated the whole myogenic program. Third, mdx-ApN mice exhibited a more oxidative and resistant myofiber (type 1) phenotype. Fourth, ApN also up-regulated utrophin, an analogue of dystrophin, which partly compensated the lack of this scaffolding protein. Eventually, mdx-ApN mice displayed higher global muscular force and endurance along with decreased muscle injury. Reduction of plasma creatine kinase and lactate dehydrogenase and curtailed extravasation of *i.p.* dye solution into muscles mirrored the strengthening of sarcolemmal integrity [56]. All these beneficial effects of ApN have recently been reproduced by AdipoRon, an agonist of ApN receptor given orally for 2 months to young mdx mice [103] (see Section 6.3).

As expected, a reverse pattern was observed with an *a contrario* approach. When mdx mice were crossed with ApN-knockout mice, the mdx phenotype actually worsened. These mdx mice exhibited lower global muscle force and endurance as well as increased muscle damage when compared to regular mdx mice. However, ApN supplementation by muscular gene electro-transfer counteracted all abnormalities even once the disease has been installed, indicating that ApN can act not only as a preventive, but also as a curative agent [102].

Some of these beneficial effects of ApN/AdipoRon have been translated to humans. We found that ApN/AdipoRon supplementation retained its anti-inflammatory properties in DMD human myotubes [56,57,103] and induced a shift in the secretion of downstream myokines toward a less inflammatory profile. Indeed, secretome analysis from myotubes revealed that ApN down-regulated the secretion of two pro-inflammatory factors (TNFα and IL-17A), one soluble receptor (sTNFRII), and one chemokine (CCL28) [57].

As in healthy muscle, the mechanisms leading to the protective effects of ApN/AdipoRon in DMD involved the activation of the AdipoR1-AMPK-SIRT1-PGC-1α axis both in vivo (mdx mice) or in vitro (human dystrophic myotubes) [56,57,103] (Figure 4). Silencing each component of this cascade abrogated the effects of ApN/AdipoRon. The anti-inflammatory action of ApN/AdipoRon in DMD is crucial and has been ascribed to NF-κB inhibition by PGC-1α dephosphorylation, SIRT1 deacetylation [56] and miR-711 up-regulation [62]. PPARα could also contribute to repress NF-κB [61]. Moreover, restoration of the myogenic program could result from attenuation of inflammation [56,103] as well as from insulin-sensitizing properties or p38 MAPK activation [68,69]. Eventually, enhanced activity of PGC-1α will lead to up-regulation of several target genes contributing to the protection afforded by ApN/AdipoRon in DMD. First, antioxidant detoxifying enzymes may reduce the oxidative stress. Second, enhanced mitochondrial biogenesis and oxidative metabolism will induce a switch towards an oxidative and more resistant fiber phenotype [56,102]. Third, up-regulation of utrophin partly substitutes for dystrophin deficiency [56,57]. Fourth, up-regulation of miR-711, another PGC-1α target gene (Collagen VII alpha-1) [62] could deter inflammation by a double mechanism. Indeed, miR-711 can repress NF-κB and NLRP3 inflammasome activation [63]. NLRP3 promotes the cleavage of pro-IL-1β and pro-IL-18 leading to their active forms. NLRP3 was up-regulated in mdx mice but was then normalized in mdx-ApN mice, through miR-711 expression [63].

To date, glucocorticoids (GCs) are the only medication commonly used to manage DMD. However, their chronic use is hindered by several side effects, some of which involving cushingoid appearance, muscle atrophy, glucose intolerance, weight gain, hypertension and behavior disorders [104]. ApN/AdipoRon could well be strong alternatives to GCs due to their pleomorphic protective properties. These medications could even counter some GC side effects by enhancing insulin sensitivity [105,106], preventing obesity [107], hypertension [108] and mood disturbances [107].

#### 5.1.2. Other Dystrophies

The beneficial effects of ApN were also recapitulated in a different type of a hereditary muscle disorder, collagen VI-related myopathies [96]. Collagen VI-related myopathies encompass a spectrum of diseases ranging from severe Ullrich muscular dystrophy to mild Bethlem myopathy. They are caused by mutations in the three major collagen type VI genes and result in dysfunctional microfibrillar collagen VI in the extracellular matrix of muscle and other connective tissues, such as skin and tendons. In muscle, this results in dystrophic changes, fibrosis, defective autophagy and increased apoptosis [109]. Fiaschi’s team showed that *Col6a1^−/−^* (collagen VI–null) mice had decreased plasma ApN levels and that myoblasts isolated from *Col6a1^−/−^* mice had impaired secretion of ApN. These findings are reminiscent of those observed in DMD. *Col6a1^−/−^* myoblasts also displayed several metabolic features, such as impaired glucose uptake and dysfunctioning mitochondria, which were corrected by addition of exogenous ApN [96,110].

Finally, low concentration of serum ApN together with a selective reduction of its HMW oligomers were detected in patients with myotonic dystrophy type 1 (DM1), another rare genetic disorder that is characterized by muscle wasting and metabolic comorbidity including high risk of developing type 2 diabetes. It is likely that decreased ApN levels might contribute to the worsening of insulin resistance and metabolic complications in DM1 patients [111]. Yet, it is still unclear whether low ApN is the consequence of the associated comorbidities or of the muscular disease *per se*.

### 5.2. Beneficial Effects of Adiponectin on Sarcopenia

Sarcopenia is a progressive and generalized skeletal muscle disorder involving an accelerated loss of muscle mass and function that is associated with adverse health outcomes (such as falls, functional decline, frailty, and mortality) [112]. When first described, this term referred to an age-related problem, but now sarcopenia is increasingly recognized as associated with a range of long-term conditions.

Overall loss of skeletal mass results from an imbalance between muscle protein anabolic and catabolic pathways, where protein synthesis is hindered and protein breakdown is excessive [113]. Forkhead box O (FoxO) family TF play a critical role in protein breakdown by activating the expression of atrogenes (which include two muscle-specific ubiquitin ligases, atrogin-1 and MuRF1) responsible for profound loss of muscle mass [75]. The transcriptional activity of FoxO3 is inhibited, on one hand, by PGC-1α and on the other hand, by the insulin/IGF1 signaling pathway, which also activates protein synthesis [75]. Hence, ApN may deter muscle atrophy by activation of PGC-1α and its insulin-sensitizing properties. Its anti-inflammatory, anti-oxidative stress, and pro-myogenic properties may contribute as well.

Herein, we will focus on sarcopenia related to selected metabolic or endocrine disorders and on age-related sarcopenia.

#### 5.2.1. Metabolic or Endocrine Disorder-Associated Sarcopenia

Sarcopenic obesity may be linked to low circulating ApN, which prevails in obese subjects and in those with the metabolic syndrome. As yet, there is still no consensus on the definition of sarcopenic obesity [112]. However, this condition may be disregarded because of the increase in overall body weight. Low ApN levels in these subjects may thus contribute to the reduction of lean mass, together with sex-specific hormonal changes, low-grade inflammation and increased intramyocellular triglyceride content [114]. The complications resulting from obesity are additive to those of sarcopenia in these patients.

Glucocorticoids are known to promote protein catabolism and induce muscle loss. ApN has been found to mitigate steroid-induced atrophy in vitro in a murine cell line. These effects were mediated through activation of AMPK and AKT pathways that led to stimulation of PGC-1α and inhibition of FoxOs. In addition, ApN counteracted muscle atrophy in rats challenged in vivo by glucocorticoids [115].

#### 5.2.2. Age-Associated Sarcopenia

Most studies demonstrate that plasma ApN levels are higher among centenarians compared to elderly individuals. Those levels were correlated with a preferable metabolic phenotype, including good lipid profile and insulin sensitivity, thereby signifying the beneficial metabolic effects of ApN for enhancing longevity [116]. In line with this assumption, in a “classical” population of geriatric patients, sarcopenia was associated with circulating higher inflammatory markers (CRP) and lower ApN levels [117]. However, in some life-threatening conditions associated with sarcopenia (chronic cardiac heart failure or kidney disease), ApN levels may be paradoxically elevated in patients and could represent unsuccessful compensatory mechanisms against inflammation and oxidative stress [116].

ApN treatment could counteract sarcopenia. Aged ovariectomized (OVX) rats represent an established model for postmenopausal osteopenia where sarcopenia and increased visceral fat occur concurrently. The animals were treated with ApN or ApN mimetic (GTFD; see Section 6.2) for 12 wks. Besides protection against osteopenia, ApN or its mimic reversed sarcopenia by suppressing atrogenes and stimulating MyoD while enhancing PGC-1α expression. ApN also corrected OVX-induced gain in body weight and changes in body composition (lean and fat mass), as well as insulin resistance [118].

Restoration of the balance between muscle apoptosis and proliferation may account for the prevention of age-related muscle dysfunction and wasting. A study conducted in a mouse model of accelerated senescence reported that ApN mediates the stimulatory effects of exercise on muscle stem cell proliferation, while mitigating apoptosis. These changes were associated with improved exercise force and endurance in these aged mice and resulted from activation of AdipoR1-AMPK axis [80].

## 6. Adiponectin Mimics

The direct use of ApN as a therapeutic agent has several challenges. ApN possesses complex three-dimensional structures, it circulates at high concentration in plasma and, as with any other peptide, it must be injected. Developing a new treatment mimicking the beneficial effects of ApN and that can be easily produced and administered is thus of great interest [119]. Several short peptides and small molecules have been identified today to target AdipoRs and mimic some of ApN effects. These receptor agonists can induce similar downstream signaling cascades to ApN, such as AMPK, p38MAPK, and PPARα [120] (Table 1).

### 6.1. Peptides and Proteins

ADP355 is a short peptide with a sequence resembling the C-terminal globular domain of ApN and thus with a biological activity resembling gApN. ADP355 was shown to restrict the proliferation of cancer lines in vitro and suppress the growth of human breast cancer xenografts in mice in vivo [121]. After this initial success and since ApN circulates in multimeric forms, a second generation of the peptide (ADP399) was formed. The dimeric ADP399 exhibited a 20-fold increase in activity when compared to the monomeric ADP355 [122].

ADP-1 is a highly conserved 13-residue segment peptide from ApN’s collagen domain, with similar positive effects on glucose and fatty-acid metabolism. It was shown to increase glucose uptake in rat skeletal muscle cells lines and improved glucose tolerance in *db/db* mice [123].

Osmotin is a plant protein possessing structural and functional similarities to ApN, and with AdipoR1 being the mammalian homologue of the osmotin receptor [124]. Several in vitro and in vivo studies revealed its potential effects for counteracting obesity, diabetes and cardiovascular diseases [125,126,127]. Other studies have demonstrated its neuroprotective effects by preventing memory impairment and neuroinflammation induced by either amyloid or LPS injections [128,129]. It was further shown to improve the neuropathological deficits of Alzheimer’s, through an AdipoR1-AMPK-dependent mechanism, in in vitro and in vivo models of the disease [130,131], making it an interesting candidate for the treatment of neurological disorders [132].

Additional AdipoR activating peptides have also been identified in silico [120]. Pep70 is a potential AdipoR1 agonist and could significantly inhibit the proliferation of cells and suppress the expression of collagen type I alpha1 and TGF-β1, thus protecting against fibrotic responses [133]. PEGylated BHD1028 is an AdipoR1 agonist and was proposed for the treatment of diabetes [134].

### 6.2. Flavonoids

GTDF is a flavonoid-type compound, a naturally occurring quercetin analogue, which was shown to attenuate the diabetic phenotype and its complications in obese *db/db* mice, through an AdipoR1-dependent mechanism [135]. GTDF also protected against skeletal muscle atrophy, by increasing myoblast differentiation and suppressing the expressions of atrophy markers, such as atrogin-1 and MuRF1, in models of steroid, cytokine and starvation-induced muscle atrophy [115].

Tiliroside is a plant derived glycosidic flavonoid with ApN-like effect. It was shown to stimulate fatty-acid oxidation in liver and skeletal muscle in obese-diabetic mice, and attenuate obesity-induced metabolic disorders [136].

### 6.3. AdipoRon

Last but not least, the most compelling mimic to date is AdipoRon, an orally active synthetic small-molecule AdipoRs agonist identified in 2013 by Okada-Iwabu et al. after screening a compound library to identify those that bind to AdipoRs and greatly activate AMPK [106]. AdipoRon has first been proposed for the treatment of type 2 diabetes and other obesity-related disorders in mice [32,106,107,108,137,138]. It could even prolong the lifespan of obese treated mice [105,106]. Since then, its beneficial effects have also been demonstrated in other non-metabolic conditions such as cancer [139,140], systemic sclerosis [141], depression [107], anxiety [142], post-traumatic stress disorder [143], and cerebral ischemia [144,145]. Our latest study has brought to light the beneficial and protective effects of AdipoRon on the dystrophic skeletal muscle in mdx mice and in human DMD myotubes. Indeed, AdipoRon, mainly though AdipoR1-AMPK signaling, significantly mitigated skeletal muscle inflammation, oxidative stress, and damage, while markedly improving its regeneration and function [103].

## 7. Conclusions

ApN is a powerful hormone that exerts various beneficial effects on different organs and shares similar signaling pathways to exercise. It protects muscle against metabolic disturbances, inflammation, and oxidative stress, and favors regeneration, even when abnormalities are extremely severe such as in dystrophinopathies. Development of small AdipoRs agonists is of great interest and represents an important step towards filling an unmet clinical need for additional therapeutic options to potentially prevent or combat not only skeletal muscle disorders but also a plethora of other diseases with sterile inflammation or metabolic dysfunction.

## Figures and Tables

**Figure 1 ijms-21-02620-f001:**
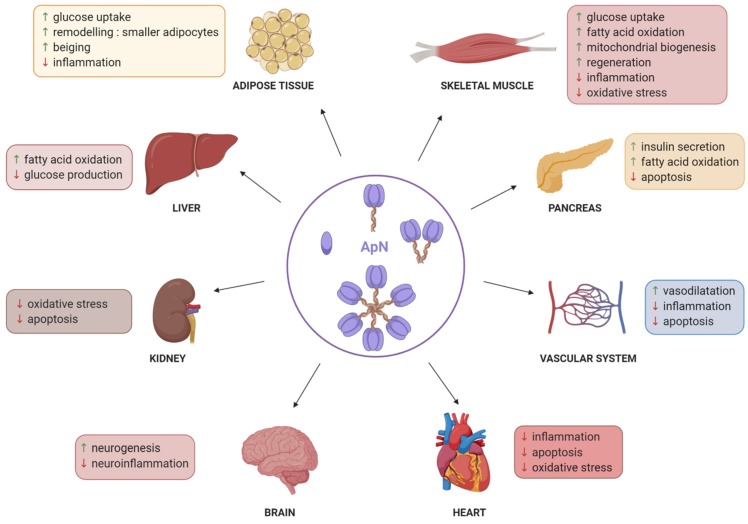
Pleiotropic effects of adiponectin. This figure summarizes the beneficial and protective effects of ApN on a variety of tissues and organs. Green arrows represent an increase, while red arrows represent a decrease. ApN, adiponectin as globular, trimeric, hexameric or multimeric forms (starting from left symbol and going in a clockwise direction). Created with BioRender.com.

**Figure 2 ijms-21-02620-f002:**
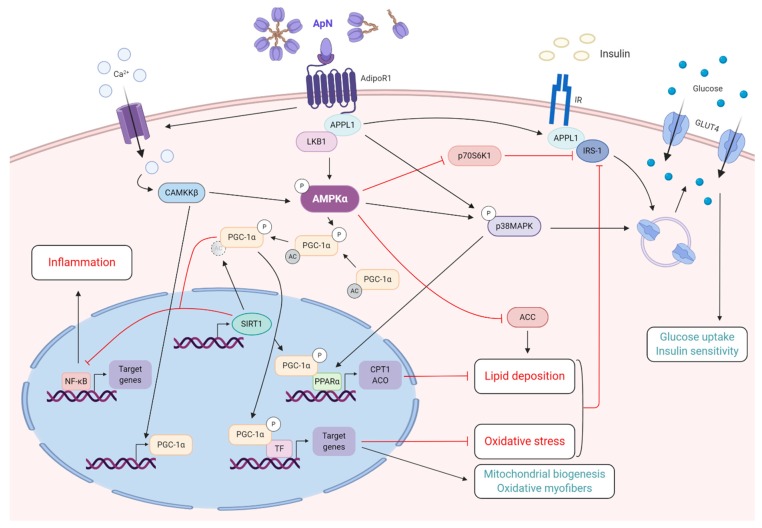
Adiponectin properties on healthy skeletal muscle. This figure summarizes the main metabolic effects of ApN, while specifically highlighting a central crosstalk between ApN-AdipoR1-AMPK and insulin pathways. Briefly, binding of ApN to AdipoR1 will recruit LKB1 and increase calcium influx, both required to fully activate AMPK-SIRT1-PGC-1α axis. Then, AMPK signaling will repress the activity of NF-κB and decrease muscle inflammation. It will also increase, via PGC-1α, mitochondrial biogenesis and function and favors an oxidative myofiber phenotype, while markedly decreasing oxidative stress. In addition, this pathway will limit and reduce lipid deposition via direct action, through inhibition of ACC, or indirect one, through activation of PPARα and its target genes (*CPT1* and *ACO*). Eventually, decrease in muscle oxidative stress and lipid content, as well as inhibition of p70S6K1, will significantly reduce insulin resistance. Finally, ApN could also boost insulin sensitivity and glucose uptake by promoting IRS1 binding to insulin receptor and by activating p38MAPK, respectively. Pointed head black arrows indicate activation or induction, while blunt head red arrows indicate inhibition. Dashed and blurred circle represents removal of the indicated residue. Boxes with processes in green represent net beneficial effects of ApN, while boxes with processes in red represent metabolic dysfunctions counteracted by ApN. ApN, adiponectin. TF, transcription factors. Created with BioRender.com.

**Figure 3 ijms-21-02620-f003:**
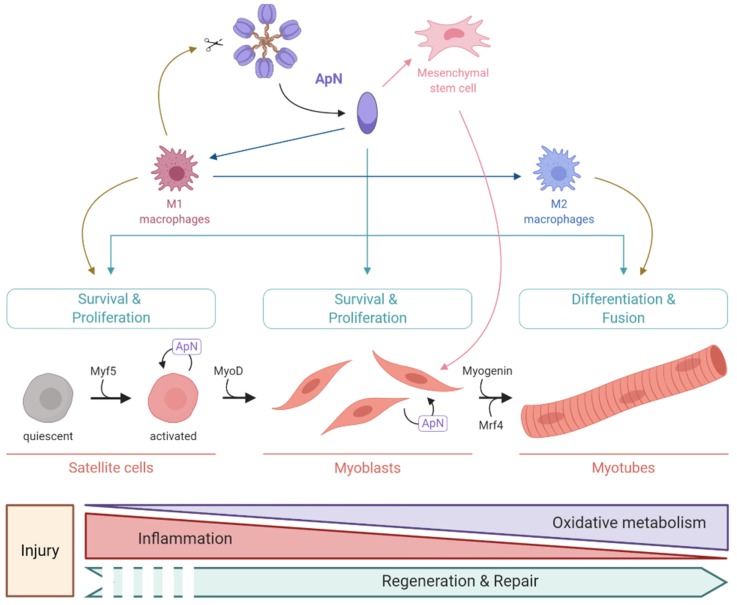
Pro-myogenic properties of ApN. This figure summarizes the multiple beneficial effects of ApN on several steps of muscle regeneration (boxes in green), while highlighting an important interplay among 3 major processes, including resolution of inflammation, energy metabolism, and muscle repair. These pro-myogenic properties of ApN have only been tested on rodent skeletal muscle thus far. Green arrows represent the direct pro-myogenic effects of ApN. Blue arrows represent an indirect pro-myogenic effect of ApN by inducing a shift from pro-inflammatory M1 to anti-inflammatory M2 macrophages, for a resolution of inflammation and activation of the tissue-healing process. Pink arrows represent an indirect pro-myogenic effect of ApN by stimulating non-muscle stem cells and driving their commitment towards the skeletal muscle lineage. Brown arrows represent the different effects of macrophages. Created with BioRender.com.

**Figure 4 ijms-21-02620-f004:**
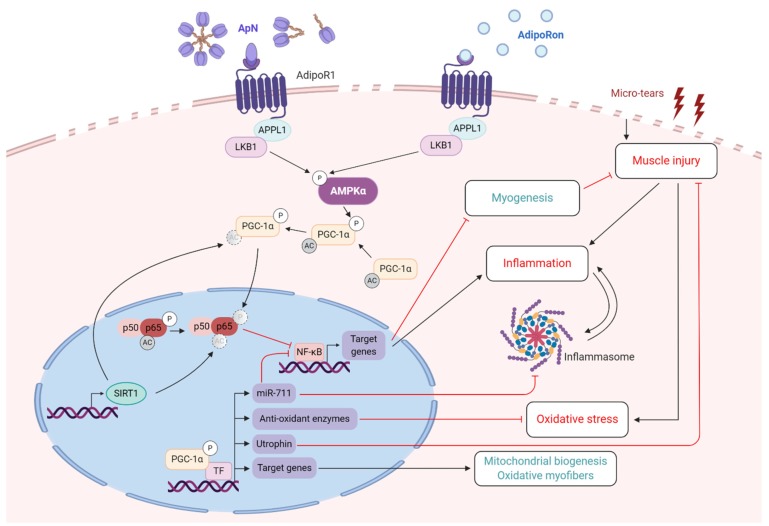
Adiponectin/AdipoRon properties on diseased skeletal muscle. This figure summarizes the beneficial and protective effects of ApN and AdipoRon on dystrophic skeletal muscle, which is characterized by micro-tears in the sarcolemmal membrane due to lack of dystrophin protein. It also highlights their mechanisms of actions. Briefly, binding of either ApN or AdipoRon to AdipoR1 will activate AMPK-SIRT1-PGC-1α pathway. Then, PGC-1α represses NF-κB activity by de-phosphorylation of the p65 subunit, while SIRT1 represses it by deacetylation. This results in reduction of inflammation and an improved myogenic program. In addition, activation of TF via PGC-1α will help mediate several beneficial effects of ApN. First, expression of different target genes will increase mitochondrial biogenesis and function and favors an oxidative and more resistant myofiber phenotype. Second, increased expression and production of utrophin, which is a dystrophin analogue, and of antioxidant enzymes, which reduce oxidative stress, will greatly protect the dystrophic muscle against excessive damage. Third, up-regulation of miR-711 will significantly hinder inflammasome priming and activation, and further repress NF-κB activity, thus helping to put a brake on muscle inflammation. Thus far, these properties have been tested on skeletal muscle from mdx mice and confirmed in human DMD myotubes. Pointed head black arrows indicate activation or induction, while blunt head red arrows indicate inhibition. Dashed and blurred circles represent removal of the indicated residue. Boxes with processes in green represent net beneficial effects of ApN, while boxes with processes in red represent deleterious factors inhibited by ApN. ApN, adiponectin. TF, transcription factors. Created with BioRender.com.

**Table 1 ijms-21-02620-t001:** Adiponectin mimics. Compounds that specifically bind to AdipoRs and greatly mimic several beneficial and protective effects of ApN.

Compound	Nature	Effects	Signaling	References
ADP355/ADP399	Short peptides	Suppress cancer cells proliferation	AdipoR1- AMPK, AKT, STAT3, ERK1/2	Otvos et al. 2011
ADP-1	Short peptide	Glucose and lipid metabolisms	AdipoR1-AMPK	Sayed et al. 2018
Osmotin	Protein	Pro-metabolic and neuroprotective properties	AdipoR1-AMPK	Anil et al. 2015 Ali et al. 2015 Shah et al. 2017
Pep70	Peptide	Anti-fibrosis action	AdipoR1	Ma et al. 2017
PEGylated BHD1028	Peptide	Anti-diabetic action	AdipoR1	Kim et al. 2018
GTDF	Flavonoid	Pro-metabolic and anti-atrophic effects	AdipoR1-AMPK, AKT	Singh et al. 2014, 2017
Tiliroside	Flavonoid	Lipid metabolism	AdipoR1-AMPK AdipoR2-PPARα	Goto et al. 2012
AdipoRon	Small synthetic molecule	Pleiotropic effects similar to ApN	AdipoR1-AMPK AdipoR2-PPARα	Okada-Iwabu et al. 2013 Abou-Samra et al. 2020

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
