# Peer review of "Adiponectin and Its Mimics on Skeletal Muscle: Insulin Sensitizers, Fat Burners, Exercise Mimickers, Muscling Pills … or Everything Together?"

_ijms, 2020, doi:10.3390/ijms21072620_

Round 1
Reviewer 1 Report
The review “Adiponectin and its mimics on skeletal muscle: insulin-sensitizers, fat-burners, exercise mimickers, muscling pills … or everything together?” by Michel Abou-Samra is very interesting and represents a broad and comprehensive overview of the multiple effects of adiponectin on skeletal muscle. In addition, the work also deals with a series of adiponectin mimetics of great interest and application potential. Overall, the review is well organized and clearly written; only minimal revisions are required
Concerning paragraph 2 “Adiponectin induction by skeletal muscle”
I suggest to replace “induction “ with “production” in the title of the paragraph or in alternative “induction in skeletal…”
Concerning paragraph 4.1 “Effect of exercise on ApN parameters”
The authors cited relevant studies on animal models (mice and rat), but disregarded several studies performed on humans. The effect of exercise on plasma ApN level was investigated in obese subjects and healthy subjects both trained and untrained. A summary of more relevant results should be included. Furthermore, the authors could also cite articles linking adiponectin expression level to physical activity behaviour, since these studies are more consistent although differences depending on age and gender should be considered
Author Response
Reviewer #1:
- Concerning paragraph 2 “Adiponectin induction by skeletal muscle”
I suggest to replace “induction “ with “production” in the title of the paragraph or in alternative “induction in skeletal…”. Has been done
- Concerning paragraph 4.1 “Effect of exercise on ApN parameters”
The authors could also cite articles linking adiponectin expression level to physical activity behaviour.
We have now added two sentences on adiponectin level and physical activity behaviour with 2 new references (86 & 87). Because this point has been extensively addressed in another review, we mention this recent review (ref 78) for readers willing to have more information on the topic.
We added 2 sentences at the end of p. 7
Reviewer 2 Report
The review “Adiponectin and its mimics on skeletal muscle: insulin-sensitizers, fatburners, exercise mimickers, muscling pills … or everything together?” by Michel Abou-Samra et al. reports all the beneficial and protective properties that ApN can exert, specifically on the skeletal muscle as a target tissue. We will highlight its effects and mechanisms of action, first in healthy skeletal muscle including exercised muscle, and second in diseased muscle from a variety of pathological conditions. In fact, they detailes much of the information on the actions carried out by adiponectin even if it does not lead to a useful conclusion on what is the pre-eminent role of this adipokine.
It is no clearly understanding when the author refers to Human and when to the animal model. Sometimes it reports data without specifying whether it derives from experiments carried out on an animal or human model.
Furthermore, in the part concerning skeletal muscle and the effect of ApN (Adiponectin and exercise) they do not reported recent pubblications and therefore the recent bibliography is also missing, concerning the involvement of inflammation and peripheral signals (among them ApN) in oxidative stress caused by sports training.
Moreover in the chapter on metabolism of skeletal muscles they do not report some recent results obtained on the role of adiponectin in the in collagen VI-related myopathies.
For this reason, I believe that the review cannot be accepted in this form.
I propose a review considering these points:
1- better distinguish the information obtained from animal models or from human studies
2- add recent data regarding the involvement of inflammation and peripheral signals (among them ApN) in oxidative stress caused by exercise training
3- report some recent results obtained on the role of adiponectin in the in collagen VI-related myopathies.
Author Response
Reviewer #2:
General comment
The authors detail much of the information on the actions carried out by adiponectin even if it does not lead to a useful conclusion on what is the pre-eminent role of this adipokine.
There is an interplay among the 3 major functions of ApN (metabolic, anti-inflammatory and pro-myogenic properties) in skeletal muscle. These 3 functions are equally important and cooperate together as mentioned in the last para of our MS dealing with pro-myogenic effects on p. 7 (“There is an interplay …) and further illustrated at the bottom of Fig. 3.
Specific comments
- Better distinguish the information obtained from animal models or from human studies
We have added two sentences in the introduction of Chap 3 dealing with “Adiponectin properties on healthy skeletal muscle” to indicate that in vivo data have only been obtained in rodents and that most mouse data have been confirmed in human myotubes in vitro.
Two sentences have been added in Chap. 3 p. 4 to mention this fact. Moreover, on several occasions, we have now specified when experiments were performed in mice or humans. Eventually, we have also indicated in the legend of Fig. 3 one sentence underlined in yellow to specify that “These pro-myogenic properties of ApN have only been tested on rodent skeletal muscle thus far”, and another in the legend of Fig. 4 “Thus far, these properties have been tested on skeletal muscle from mdx mice and confirmed in human DMD myotubes”.
- Add recent data regarding the involvement of inflammation and peripheral signals (among them ApN) in oxidative stress caused by exercise training
A new sentence, along with a new ref 91, have been added at the end of the para “4.2 Contribution of ApN to exercise-induced improvement in metabolic fitness”, to mention these recent data
- Report some recent results obtained on the role of adiponectin in the in collagen VI-related myopathies.
Section 5.1.b on p. 10 is dedicated to this subject, and it includes the most recent results obtained. However, references have now been corrected and updated. Ref 97 has been added, and the previous ref 94 has become 111.